# Workplace Wellbeing and Quality of Life Perceived by Portuguese Nurses during the COVID-19 Pandemic: The Role of Protective Factors and Stressors

**DOI:** 10.3390/ijerph192114231

**Published:** 2022-10-31

**Authors:** Francisco Sampaio, Ricardo Salgado, Matteo Antonini, Philippe Delmas, Annie Oulevey Bachmann, Ingrid Gilles, Claudia Ortoleva Bucher

**Affiliations:** 1Higher School of Health Fernando Pessoa, Rua Delfim Maia, 334, 4200-253 Porto, Portugal; 2CINTESIS@RISE—Center for Health Technology and Services Research/Health Research Network from the Lab to the Community, Rua Dr. Plácido da Costa, 4200-450 Porto, Portugal; 3La Source School of Nursing, HES-SO University of Applied Sciences and Arts Western Switzerland, Av. Vinet 30, 1004 Lausanne, Switzerland; 4Epidemiology and Health Systems, Center for Primary Care and Public Health, 1010 Lausanne, Switzerland

**Keywords:** workplace, health, quality of life, nurses, COVID-19, pandemics, stressors, protective factors, salutogenesis

## Abstract

During the COVID-19 pandemic, nurses were exposed to many stressors, which may have been associated with some mental health problems. However, most of the studies carried out on nurses’ quality of life and workplace wellbeing during the COVID-19 pandemic took a pathogenic approach. Given that current scientific knowledge in this field presented too many gaps to properly inform preventive and therapeutic action, the aim of this study was to explore whether protective factors (resilience, perceived social support, and professional identification) and stressors (perceived stress and psychosocial risks in the workplace) influenced the quality of life and workplace wellbeing perceived by Portuguese nurses during the COVID-19 pandemic. Data for this cross-sectional study was collected through online self-administered questionnaires. Linear regression models were used to analyze the relationships between variables. Results showed that perceived stress, resilience and job satisfaction were associated with quality of life and workplace wellbeing among Portuguese nurses. The study’s findings could serve to inform health policy and should draw the attention of nursing managers to the needs and difficulties reported by nurses, to the importance of providing them with emotional support, and to the relevance of promoting a good work environment.

## 1. Introduction

The COVID-19 pandemic began in Wuhan, China, in 2019 and spread rapidly around the world. As of 29 May 2022, over 526 million confirmed cases and over six million deaths had been reported globally [1]. Over this period, nurses were exposed to different kinds of stressors, such as having to adapt to different work shifts and having to perform unusual tasks in unusual settings (e.g., units specially dedicated to the care of COVID-19 patients). This exposure may have been associated with high levels of psychological distress [2,3], anxiety, depression, and insomnia, with prevalence rates hovering around 23% [4,5]. During the Severe Acute Respiratory Syndrome (SARS) epidemic, from 17.3% to 75.3% of healthcare workers (HCW) presented mental health problems [6,7,8,9,10]. Compared with other HCWs, nurses reported higher levels of stress, posttraumatic stress, and psychopathological symptoms [11,12]. These problems persisted for as long as two years after the end of the epidemic [13]. 

Studies that took a salutogenic approach to the subject have shown that nurses can preserve their health despite a pandemic by mobilizing what is referred to as generalized resistance resources [14]. The notion of wellbeing emerged to take account of elements that people identified as specific to them and their social environment above and beyond objective elements of quality of life (QoL) related to socio-economic factors [15]. It refers to the psychological, social, and physical resources that individuals use to deal with life challenges [16] and to the fact that individuals can have agency in their social environment [17]. Salutogenesis is “a scholarly orientation focusing attention on the study of the origins of health and assets for health, contra the origins of disease and risk factors” [14] (p. 7). Its particularity versus the pathogenic approach is that it considers health on a continuum running from optimal health to disease. It also seeks to highlight what it calls salutary factors or generalized health resources that actively support health, rather than taking only risk factors into account when planning preventive action [14]. Two studies have shown, respectively, that 18.4% and 56.7% of nurses did not feel stressed by the Middle East Respiratory Syndrome (MERS) [18] and SARS [19] epidemics. Recently, Lecocq et al. [20] in Belgium even suggested that the pandemic had a positive impact on nurses at both the personal and professional levels. Though nurses have not all necessarily been directly affected by the COVID-19 pandemic, those who have managed to mobilize different types of resources (referred to, from a salutogenic perspective, as “generalized resistance resources” or “health protective factors”) were able to cope and retain their health. These resources could be personal (e.g., resilience), relational (e.g., social support), or environmental (e.g., availability of personal protective equipment) [21,22]. However, to date, there has been little research into what exactly allows nurses to retain their health. Gaining clearer insight into how nurses do this in the midst of an epidemic/pandemic is essential. This knowledge could help develop interventions aimed at maintaining their health and workplace wellbeing in pandemic times.

Two recent systematic reviews [2] regarding the current literature on COVID-19 have shown that HCWs who worked in COVID-19 units were at high risk of developing psychological symptoms [5]. However, it was not mere exposure to this stressor (i.e., having worked in a COVID-19 unit) but rather how HCWs coped with it that seemed to impact their health. According to our framework, individuals do not respond the same way to the same stressors: Some people experience or feel stress (distress) whereas others are stimulated by them in a positive manner (eustress). Consequently, we assumed that in times of crisis all HCWs, whether on the front line or not, face all sorts of stressors capable of affecting their physical and mental health. In the context of SARS, McAlonan et al. [23] showed that low-risk HCWs who worked in units that did not care for SARS patients reported levels of perceived stress just as high as high-risk HCWs who worked on the front line and that stress level was associated with severity of anxiety and depression symptoms. This also explained why HCWs with no or little direct exposure to SARS patients presented health problems later on [23]. 

As of 9 June 2022, over 4.89 million confirmed cases of COVID-19 and over 23,489 deaths had been reported in Portugal [24]. Cross-sectional and longitudinal studies [25,26] have been carried out in the country to investigate nurses’ mental health during the COVID-19 pandemic. Other studies explored the association between the mental health promotion strategies used by nurses during the COVID-19 pandemic and their symptoms of depression, anxiety, and stress [27,28]. However, none of these studies used a salutogenic approach to the phenomenon. Moreover, all focused on depression, anxiety, and stress as outcomes, and none considered factors such as resilience, perceived social support, and psychosocial risks in the workplace and their potential influence on stress, workplace wellbeing, and QoL as perceived by nurses. During the COVID-19 pandemic, in Portugal, nurses were exposed to the same kind of stressors than nurses all around the world. According to a study carried out in the country [26], the most important ones were the fear to be infected and the fear to infect others. However, few support was available for them. The exception to the lack of support were the mental health helplines, which were made available in most of the hospitals and that were operated, mainly, by clinical psychologists.

Considering that QoL and workplace wellbeing are the main outcomes of this study, it seems particularly relevant to define these concepts under the umbrella of a salutogenic model. Thus, QoL was classically defined by the World Health Organization [29] as individuals’ perceptions of their position in life in the context of the culture and value systems in which they live and in relation to their goals, expectations, standards, and concerns. On the other hand, according to the International Labour Organization [30], workplace wellbeing relates to all aspects of working life, from the quality and safety of the physical environment, to how workers feel about their work, their working environment, the climate at work and work organization. In sum, notwithstanding the above studies, current scientific knowledge presents too many gaps to properly inform preventive and therapeutic action. First, the vast majority of studies to date have adopted a pathogenic approach. Accordingly, they have focused on identifying diseases, symptoms, and risk factors [2,3]. However, as health is “a state of complete physical, mental, and social wellbeing, and not merely the absence of disease or infirmity” [29], it is essential also to assess wellbeing and to identify health resources in order to build scientific knowledge that is more comprehensive. Second, the vast majority of studies conducted to date have been atheoretical [2,3]. As a result, relationships between concepts have varied widely across studies. For example, depression has been considered both a predictor and an outcome [2,3]. A theoretical framework is essential in research to describe, explain, and understand the nature and meaning of phenomena, and in health care to develop interventions [31]. Third, how the COVID-19 crisis has been managed has differed greatly at times across countries owing to differences in political systems, economic resources, and cultural backgrounds (e.g., willingness to adhere to protection measures). Culture also plays a role in how individuals perceive stress and cope with it [32]. Thus, before transferring research results from other countries to the reality of Portuguese nurses, it seemed imperative to collect data from a sample of nurses from Portugal. Considering this rationale, we decided to conduct this study because it is the first on this topic conducted in Portugal during the COVID-19 pandemic, which aims to examine the influence of protective and stress factors on workplace well-being and QoL through the lens of salutogenesis.

Against this background, the aim of this study was to explore whether protective factors (resilience, perceived social support, and professional identification) and stressors (perceived stress and psychosocial risks in the workplace) influenced the workplace wellbeing and QoL perceived by Portuguese nurses during the COVID-19 pandemic.

## 2. Materials and Methods

### 2.1. Design and Population

This cross-sectional study was conducted in Portugal as part of a larger research project investigating nurses’ QoL and workplace wellbeing in other European countries (Switzerland, France, Belgium) and Canada [33,34]. It was carried out during the fifth COVID-19 wave from November 2021 to January 2022. Portugal’s fifth wave peaked that January and, by then, about 85% of the population was fully vaccinated with two doses [35]. In order to have as many participants as possible, an invitation containing a link to participate in an online self-administered questionnaire was posted on the website of the Portuguese Council of Nurses (Ordem dos Enfermeiros [OE]). Participation was voluntary and anonymous and targeted all nurses with clinical responsibilities in Portuguese territory. At the time, the OE totaled 80,239 members. To be eligible for the study, nurses had to meet the following inclusion criteria: engaged in clinical practice in Portugal; holding a regular work contract; and working half time or more; and fluent in Portuguese. The exclusion criteria were as follows: holding a management position; having only student status; and having only teaching status. The data collection was carried out using Sphinx IQ2 7.2.0.2. This manuscript was completed according to the Strengthening the Reporting of Observational Studies in Epidemiology (STROBE) guidelines for cross-sectional studies [36].

Against this background, we undertook a study using the Neuman Systems Model (NSM) as the theoretical framework to explore: (1) the stressors to which a population’s health is exposed; (2) the relationship between these stressors and overall health and wellbeing over time; and (3) the mechanisms used to stay healthy despite this exposure. This framework considers human beings holistically. It centers on their wellbeing and particularly on how they respond to stressors [37]. There are three main reasons that this framework seemed well suited to explore the stressors that nurses were exposed to during the COVID-19 pandemic and to examine the relationship between nurse exposure to stressors and nurse overall health: (i) it considers stressors as neutral a priori; (ii) it does not consider the relationship between stressor exposure and health status to be negative a priori; and (iii) it takes a salutogenic approach in that it facilitates identification of health protective factors when individuals are exposed to different stressors.

### 2.2. Measures

Data were collected through several assessment tools. The study outcomes—professional wellbeing and QoL—were measured using the Psychological Wellbeing Scale [38] and The World Health Organization Quality of Life—BREF (WHOQOL-BREF) [39], respectively. Stressors and protective factors were measured through the Perceived Stress Scale (PSS) [40], the 10-item Connor-Davidson Resilience Scale (CD-RISC) [41], the Multidimensional Scale of Perceived Social Support (MSPSS) [42], the Copenhagen Psychosocial Questionnaire (COPSOQ) [43], and a single item for professional identification [44]. Sociodemographic characteristics were assessed by way of a tailored sociodemographic questionnaire.

The Psychological Wellbeing Scale was developed by Diener et al. [38] and adapted by Fisher [45] for the workplace. The eight-item scale measures self-perceived functioning in areas such as self-esteem, purpose, and relationships. Items are rated on a five-point Likert scale from 1 (poor wellbeing at work) to 5 (strong wellbeing at work) [38]. This scale was translated into Portuguese following the Wild et al. [46] method. It obtained a Cronbach’s alpha of 0.83. 

The WHOQOL-BREF [39] is a 26-item scale used to measure self-perceived QoL. It covers four domains: physical health, mental wellbeing, social relations, and environment. Items are rated on a five-point scale from “not at all” to “completely”. The psychometric properties of the Portuguese version have been proven good, having obtained Cronbach’s alphas > 0.70 on all dimensions [47]. As recommended by the authors of the questionnaire, mean scores were transformed to range from 0 (poor QoL) to 100 (good QoL).

The PSS was developed by Cohen et al. [40]. Its 10-item version was translated into Portuguese and validated by Reis et al., demonstrating good psychometrics properties (Cronbach’s alpha = 0.87) [48]. The scale is used to assess how much life situations in the past month were generally perceived as threatening, unpredictable, uncontrollable, and painful [15]. The items are rated on a five-point Likert scale from 0 (never) to 4 (very often) [40].

Developed by Campbell-Sills, the CD-RISC [41] was translated and validated in Portuguese [40,49]. It serves to measure a person’s ability to bounce back when confronted with difficulties that may arise in life [41]. Items are rated on a five-point Likert scale from 0 (not at all true) to 4 (true nearly all the time) [41]. Its psychometric properties have all proved to be above 0.80 [49]. 

The MSPSS [42] was developed to measure perceived social support from family, friends, and significant others. It was translated into Portuguese [50]. Its 12 items are rated on a Likert scale from 1 (disagree completely) to 7 (agree completely). Its psychometric properties are good, with Cronbach’s alpha ranging from 0.87 to 0.95 [50].

The COPSOQ assesses psychosocial risks in the workplace through 24 core dimensions involving four aspects of work: work environment, health, wellbeing, and personality [43]. The COPSOQ has been used in various work contexts in different languages, including Portuguese [51]. For the purpose of our study, the three following dimensions were used: social support from colleagues (3 items), social support from supervisors (3 items), and job satisfaction (2 items). Items are rated on a five-point Likert scale from 1 (always) to 5 (never/hardly ever) for the two social support dimensions, and from 1 (to a very large extent) to 5 (to a very small extent) for the job satisfaction dimension. The scale has demonstrated good psychometric properties, with Cronbach’s alphas ranging from 0.75 to 0.86 for the three dimensions selected [51].

Professional identification was measured using a single-item measure of social identification from Postmes et al. [44]: “In the current context, I identify with the other nurses who do a similar job to mine.” This single item can be adapted to any social group, including professional groups [44]. The seven-point rating scale is anchored by 1 (fully disagree) and 7 (fully agree). 

The sociodemographic questionnaire was developed based on previous studies. It covered personal dimensions such as gender, age, family situation, and number of children, as well as professional dimensions, such as employment status, type of health care facility, years of experience overall, years of experience in the current department, and level of exposure to COVID-19.

### 2.3. Data Analysis

First, sociodemographic variables and variables that were assessed using assessment tools were described using absolute and relative frequencies for categorial data and means and standard deviations for numeric variables. Second, in order to assess internal consistency, Cronbach’s alphas were evaluated for instruments that were constructed from multiple items. Third, linear regression models were used to analyze the relationships between variables. For all the analyses, the significance threshold was set at *p* = 0.05 using a frequentist approach. All the analyses were conducted using R version 4.1.2 and RStudio 2022.02.0.

## 3. Results

### 3.1. Descriptive Characteristics

Of the 391 nurses who completed the questionnaires, 247 were retained for analysis after applying exclusion criteria and verifying for missing data. As shown in Table 1, 82.1% of the participants were women, 33.6% were 40 to 49 years old, 71.1% were married, and 69.6% had children. For 81.7%, more than 10 years had elapsed since obtaining their diploma; 69.6% worked in a hospital, and 93.5% had professional experience of more than five years. Since the beginning of the pandemic, 68% had been indirectly exposed to COVID-19 patients and 59% had not been reassigned to another unit.

Regarding the outcomes of this study (Table 2), participants presented a workplace wellbeing mean score of 3.74 (SD = 0.62) and QoL total mean score of 61.09 (SD = 15.55). Perceived stress scored a mean of 3.06 (SD = 0.55). Resilience obtained a mean score of 3.57 (SD = 0.66). Portuguese nurses had presented a mean score of 5.62 (SD = 1.11) when questioned about their perceived social support. The mean declared support from their colleagues and their supervisors was respectively 2.81 (SD = 1.01) and 3.95 (SD = 1.35). Job satisfaction had a mean score of 3.07 (SD = 1.29). Lastly, professional identification was evaluated with a mean score of 4.98 (SD = 1.29).

### 3.2. Regression Analysis

Two multiple linear regressions were run directly on the full sample. The first had QoL mean total score (Table 3) as the outcome variable, and the second, workplace wellbeing mean score (Table 4). The following stressors and protective factors were entered as independent variables: perceived stress, social support, resilience, psychosocial risks, professional identification, and the sociodemographic variables. The adjusted R^2^ values for the regression models were 0.67 and 0.53, respectively.

The models revealed that participants with a higher level of perceived stress had lower QoL (β = −10.33; *p* < 0.01) and lower workplace wellbeing (β = −0.23; *p* < 0.01). Perceived social support was positively associated with higher QoL (β = 4.31; *p* < 0.01) but was not significantly associated with workplace wellbeing (β = −0.01; *p* > 0.05). Resilience, too, was positively associated with QoL (β = 3.55; *p* < 0.01) and workplace wellbeing (β = 0.17; *p* < 0.01). Higher self-reported social support from colleagues and higher self-reported job satisfaction were also positively associated with both higher QoL (β = −2.40, *p* < 0.01 and β = −3.62, *p* < 0.01, respectively) and workplace wellbeing (β = −0.15, *p* < 0.01 and β = −0.21, *p* < 0.01, respectively).

## 4. Discussion

The aim of our study was to explore whether protective factors (resilience, social support, and professional identification) and stressors (stress and psychosocial risks in the workplace) influenced the workplace wellbeing and QoL perceived by Portuguese nurses during the COVID-19 pandemic. Our findings show that perceived stress led to poorer QoL and workplace wellbeing. On the other hand, resilience and job satisfaction proved to be protective factors, that is, nurses who presented higher levels of resilience and job satisfaction also showed better QoL and workplace wellbeing. Finally, nurses who had more perceived social support also presented better QoL.

Our study sample was composed of 247 nurses. Of these, 82.1% were women, compared with 82.4% within the entire nurse population in Portugal. However, whereas 31–40 age group is the largest among Portuguese nurses as a whole, 40- to 49-year-olds were the most prevalent in our sample. In other words, our study sample was a little older than the overall nurse population in Portugal. Finally, more than 50% of nurses in Portugal work in hospitals [52], compared with 69.6% in our sample. Moreover, the fact that we used non-probability sampling means that we cannot assume that our sample was representative. However, while we did not seek representativeness when collecting data through an online, self-administered questionnaire, there were some similarities between our sample and the general nurse population in Portugal.

Regarding nurses’ QoL during the COVID-19 pandemic, our study identified stress as a factor strongly associated with lower QoL. This is in line with previous studies, such as the one conducted by Yan et al. [53] in hospitals specialized in infectious diseases and in infectious disease departments at general hospitals in China from January to July 2021 (during the COVID-19 pandemic), which found that occupational stress had a major impact on nurses’ QoL. This same study [53] also found that psychological resilience played a partial role in that impact, which is also consistent with our findings. In fact, our study and the one by Yan et al. present similar findings despite having been conducted in two substantially different geographical and cultural settings (Portugal and China). This suggests that these types of psychological phenomena tend to follow similar trends in different countries and contexts. In another study, Peñacoba et al. [54], too, found stress to be associated with the physical and mental health components of QoL among intensive care nurses during the COVID-19 pandemic.

Our findings highlighted perceived social support (including from colleagues), resilience and job satisfaction as protective factors associated with better QoL during the COVID-19 pandemic. This is in line with the findings of a study conducted in Spain (a country geographically and culturally similar to Portugal) in 2018 (prior to the COVID-19 pandemic) with a sample of emergency room nurses, where perceived social support proved significantly related to all the dimensions of professional QoL [55]. In this regard, a cross-sectional study recently carried out with a sample of nurses working in a university hospital in Turkey (during the COVID-19 pandemic) observed a positive and linear correlation between job satisfaction and QoL, meaning that as nurses’ job satisfaction increases, so does their QoL [56].

Our findings also showed perceived stress to be a factor associated with a lower workplace wellbeing. Research on nurses’ wellbeing is strongly skewed toward nurses’ experiences of psychological phenomena, such as stress [57,58,59]. However, no study has ever clearly linked stress and workplace wellbeing. Nonetheless, though a direct and clear relationship between the two variables has never been established, in 2014 Karimi et al. [60] underscored that stress-related presenteeism significantly predicted nurses’ wellbeing.

In addition, our study also identified resilience, job satisfaction, and perceived social support from colleagues as protective factors associated with better nurses’ workplace wellbeing during the COVID-19 pandemic. Of these three factors, job satisfaction was the most significant. This is supported by findings of a study conducted by Chung et al. in Taiwan (prior to the COVID-19 pandemic), where a moderate positive correlation was observed between nurses’ work environment and their sense of wellbeing [61]. It is important to point out, however, that wellbeing in this other study was defined as a person’s perception of influential factors in the surroundings, and not only workplace wellbeing. Where resilience is concerned, our results seem to be in line with those of a study conducted in Australia, in 2020 (prior to the COVID-19 pandemic), which showed workplace resilience to be positively related to psychological wellbeing across all stressor categories (consumer/carer, colleague, organizational role, and organization service) [62]. Finally, the positive association we observed between perceived social support from colleagues and workplace wellbeing is consistent with several literature reviews (conducted prior to the COVID-19 pandemic) that found social support, whether provided by superiors or colleagues, to be a predictor of job satisfaction, work involvement, and nurse commitment to the organization [63,64].

The main strength of our study is that we applied a salutogenic theoretical approach that considered not only stressors but also protective factors that potentially influenced nurses’ QoL and workplace wellbeing during the COVID-19 pandemic. Nonetheless, considering that few studies carried out on this topic used a salutogenic approach, there is still a need to better fill this research gap. Moreover, it took account of several variables (stressors and protective factors), which made it possible to estimate the degree of change in the outcome variables for every unit of change in the explanatory variables. Finally, workplace wellbeing has seldom been considered as an outcome variable in the literature on nurses’ overall health during the COVID-19 pandemic.

Our study also has limitations that need to be addressed clearly. The first has to do with the sampling method. Given that the invitation to participate in the study with the link to the online self-administered questionnaires was posted on the OE website, it was not possible for us to calculate the response rate for lack of information on how many nurses accessed the website or to ensure that the sample was representative of the nurse population in Portugal. The second limitation regards the sample size. Though the sample was not small, our findings would certainly have been more robust with a larger one. The large number of online questionnaires that nurses were asked to fill out during the trying times of a pandemic might explain why we had trouble recruiting more participants. Another factor that might have affected our findings is the data collection period. As Portugal had already achieved excellent COVID-19 vaccination coverage by then (November 2021 to January 2022) [35], the psychological impact of the pandemic on nurses had probably moderated by then. Finally, our study design itself could also be considered a limitation in that cross-sectional studies measure the possible causes (stressors and protective factors) and effects (QoL and workplace wellbeing) at the same point in time.

## 5. Conclusions

This study highlighted several factors, such as the resilience, job satisfaction, and perceived social support (including from colleagues), which play a key role as protective factors of nurses’ mental health even during a crisis such as the COVID-19 pandemic. That is, probably, the most important finding from this study, which brings to science a more salutogenic approach to this phenomenon. 

The findings from this study could be used to inform health policy and nursing management. Regarding the former, they should draw the attention of decision-makers to the vital importance of designing and implementing health promotion policies for nurses specifically targeting their mental health, and of taking preventive measures that could be useful in future in the face of potential new disease outbreaks. As for nursing management, our findings should draw the attention of nursing managers to the needs and difficulties reported by nurses, to the importance of providing them with emotional support, and to the relevance of promoting a good work environment. Nonetheless, it would be vital to repeat the data collection with the same sample over time, as it could allow us to clearly identify risk factors for low QoL and/or workplace wellbeing among nurses. 

## Figures and Tables

**Table 1 ijerph-19-14231-t001:** Sociodemographic variables.

Sociodemographic Variables	n (%)
Gender	
Male	44 (17.9)
Female	202 (82.1)
Other	0 (0)
Age (in years)	
18–29	26 (10.5)
30–39	75 (30.4)
40–49	83 (33.6)
50+	63 (25.5)
Family situation	
Single	52 (21.2)
Married	175 (71.1)
Other	19 (7.7)
Children	
Yes	167 (69.6)
No	73 (30.4)
Health care facility	
Hospital	172 (69.6)
Private clinic	12 (4.9)
Rehabilitation centre	1 (0.4)
Community health centre	60 (24.3)
Retirement home	11 (4.5)
Home care	13 (5.3)
Time elapsed since diploma obtained	
<5 years	15 (6.1)
5–10 years	30 (12.2)
>10 years	201 (81.7)
Years’ experience in current department	
<2 years	1 (0.4)
2–5 years	15 (6.1)
>5 years	230 (93.5)
Exposure to COVID-19	
None	0 (0)
Direct	79 (32)
Indirect	168 (68)
Unit reassignment	
Yes	100 (41)
No	144 (59)

**Table 2 ijerph-19-14231-t002:** Outcome, stressor, and protective variable scores.

Outcomes, Stressors, and Protective Variables	Mean (SD)
	(N = 247)
**Outcomes**	
Quality of life	61.09 (15.55)
Workplace wellbeing	3.74 (0.62)
**Stressors/protective variables**	
Perceived stress	3.06 (0.55)
Perceived social support	5.62 (1.11)
Resilience	3.57 (0.66)
Social support from colleagues	2.81 (1.01) •
Social support from supervisors	3.95 (1.35) •
Job satisfaction	3.07 (0.93) •
Professional identification	4.98 (1.29)

•: inverted score as mentioned in Section 2.2. SD: standard deviation.

**Table 3 ijerph-19-14231-t003:** Association between stressors/protective factors and quality of life (regression model).

Variables	β	Std. Error	*p* Value *
Perceived stress	−10.33	1.57	<0.01 *
Perceived social support	4.31	0.68	<0.01 *
Resilience	3.55	1.12	<0.01 *
Social support from colleagues	−2.40 •	0.82	<0.01 *
Social support from supervisors	1.01 •	0.56	0.08
Job satisfaction	−3.62 •	0.87	<0.01 *
Professional identification	−0.50	0.61	0.41
Gender			
Men	2.8	1.79	0.12
Age (in years)			
30–39	0.40	3.85	0.92
40–49	−2.08	4.19	0.62
50+	−1.14	4.30	0.79
Family situation			
Married	−2.67	2.13	0.21
Other	−2.26	3.48	0.52
Children			
No	2.16	2.04	0.29
Health care facility			
Hospital	0.92	2.56	0.72
Private clinic	0.43	3.19	0.89
Community health center	−0.37	2.88	0.91
Retirement home	−4.64	3.32	0.16
Home care	0.43	2.97	0.87
Time elapsed since diploma obtained			
5–10 years	−6.88	4.27	0.11
>10 years	−5.78	4.03	0.15
Years’ experience in current department			
2–5 years	9.20	10.27	0.37
>5 years	13.95	11.31	0.22
Exposure to COVID-19			
Indirect	−0.29	1.46	0.84
Unit reassignment			
Yes	−0.13	1.33	0.92
Adjusted R^2^	0.67		

* *p* < 0.05. • inverted score as mentioned in Section 2.2. β: estimate. Std Error: Standard Error.

**Table 4 ijerph-19-14231-t004:** Association between stressors/protective factors and workplace wellbeing.

Variables	β	Std. Error	*p* Value *
Perceived stress	−0.30	0.07	<0.01 *
Perceived social support	−0.01	0.03	0.87
Resilience	0.17	0.05	<0.01 *
Social support from colleagues	−0.15 •	0.04	<0.01 *
Social support from supervisors	<0.01 •	0.03	0.91
Job satisfaction	−0.21 •	0.04	<0.01 *
Professional identification	0.02	0.03	0.48
Gender			
Men	−0.09	0.09	0.32
Age (in years)			
30–39	−0.01	0.19	0.96
40–49	−0.07	0.20	0.74
50+	−0.01	0.21	0.97
Family situation			
Married	0.11	0.10	0.28
Other	0.44	0.16	0.01 *
Children			
Yes	−0.01	0.10	0.92
Health care facility			
Hospital	−0.09	0.12	0.47
Private clinic	−0.02	0.15	0.91
Community health center	0.03	0.14	0.84
Retirement home	−0.18	0.16	0.26
Home care	−0.19	0.14	0.19
Time elapsed since diploma obtained			
5–10 years	−0.15	0.21	0.47
>10 years	−0.07	0.19	0.72
Years’ experience in current department			
2–5 years	0.05	0.50	0.92
>5 years	0.01	0.55	0.99
Exposure to COVID-19			
Indirect	−0.01	0.07	0.93
Adjusted R^2^	0.53		

* *p* < 0.05. •: inverted score as mentioned in Section 2.2. β: estimate. Std Error: Standard Error.

## Data Availability

The data presented in this study are available on request from the corresponding author. The data are not publicly available owing to the presence of health information.

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
