# Peer review of "Workplace Wellbeing and Quality of Life Perceived by Portuguese Nurses during the COVID-19 Pandemic: The Role of Protective Factors and Stressors"

_ijerph, 2022, doi:10.3390/ijerph192114231_

Round 1
Reviewer 1 Report
Thank you for inviting me to review the article entitled “Workplace Wellbeing and Quality of Life Perceived by Portuguese Nurses During the COVID-19 Pandemic: The Role of Protective Factors and Stressors”. This study aims to explore the relationship between factors such as resilience, social or professional support and perceived stress factors or psychosocial risks in the workplace and the quality of life and well-being in the workplace as perceived by Portuguese nurses during the COVID-19 pandemic. In the following, I will make some comments that I hope will be useful to the authors:
1. In the introductory section, the authors should briefly expand and justify the concept of quality of life and professional well-being under the umbrella of a salutogenic model, since these will be the variables addressed as the main outcomes of the study.
2. To facilitate the reader's understanding in the results section, it should be made clear that the dimensions social support from colleagues, social support from supervisor, and job satisfaction have an inverse interpretation to the others (although the authors already mention this adequately in the methods section).
3. In the discussion section, and thinking about the implications of the results for the post-pandemic period, it would be good to highlight the comparison of results with studies carried out during the pandemic period and the comparison of results with pre-pandemic studies.
Author Response
Dear reviewer,
Thank you very much for your relevant recommendations. We are sure they helped improved the overall quality of the paper. You can find attached a point-by-point response to all your comments.

Reviewer 2 Report
Dear Authors, before publication, the manuscript needs to be improved. Details in the attachment.

Author Response

(The authors gave the same response as above.)

Round 2
Reviewer 2 Report
Accept in present form.